# Position: Bridge the AI development-regulation gap through dedicated committees and adaptive legislation

**Mansur Ali Khan** [1]   **Mehmet Efe Akengin** [2]   **Osman Salahuddin** [3]   **Ahmad A. Rushdi** [4]

## Abstract

While AI models advance at unprecedented rates, AI safety legislation in the United States remains largely stalled or unrealized. We observe that AI policy activity is increasing globally, yet binding enactments remain limited relative to the pace of technical capability releases. **We argue that there is a need to actively bridge the AI development-regulation gap, and this requires dedicated AI committees, adaptable and preemptive legislation that is informed by comprehensive stakeholder input.** We support our position through a technical analysis of all U.S. AI-related bills introduced from 2017 to 2025, showing that only 4.23% of U.S. AI bills reach any terminal outcome (6.25% for general bills). We identify a stage-specific pattern of legislative stalling: public-interest topics such as Deepfakes and Job Security predict initial committee engagement, while structural factors including the number of bill sponsors and chamber of origin predict late-stage advancement failure. Our comprehensive analysis of institutional, economic, political, and informational constraints shows factors exacerbating these regulatory delays. To address this multifaceted gap, we propose policy recommendations grounded in planned adaptation, preemptive enactment, and independent AI oversight. Finally, we highlight the need for coordinated action across policymakers, developers, and industry stakeholders so that AI safety governance keeps pace with technological innovation.

[1]University of Washington, Bothell, WA, USA [2]Fatima Institute for Global AI Research [3]Washington State Legislature, Olympia, WA, USA [4]Stanford Institute for Human-Centered AI, Stanford University, Stanford, CA, USA. Correspondence to: Mansur Ali Khan <mkhan30@uw.edu>, Ahmad A. Rushdi <rushdi@stanford.edu>.

*Proceedings of the 43rd International Conference on Machine Learning*, Seoul, South Korea. PMLR 306, 2026. Copyright 2026 by the author(s).

## 1. Introduction

*Table 1.* Summary of Current AI Governance Approaches Across Major Regions

| Region | Primary Approach | Binding Authority | Current Limitations |
|---|---|---|---|
| European Union | Comprehensive ex ante regulation (AI Act) | Yes | Deferred enforcement details; limited individual redress |
| Asia-Pacific | Principle-based guidelines | Mostly No | Voluntary adoption; uneven national coverage |
| United States | Fragmented federal and state actions | Limited | Low federal enactment; committee-level stalling |

Our position is that **the gap between AI development and regulation must be bridged, and doing so requires dedicated committees and adaptive legislation so that stakeholders collectively close the governance gap**. Closing this gap would help align AI development with societal values, thereby increasing public trust and safety of AI systems. However, if this gap is not closed in a timely or comprehensive manner, society faces heightened risks, including but not limited to the recently studied risks of increased LLM-mediated self-harm ideation (Klein, 2025), access to AI tools capable of generating bio-weapon blueprints (Gertrude Hattoh & Akogo, 2025), and the proliferation of hyper-addictive "digital heroin" media (Khraishi et al., 2025).

Artificial Intelligence (AI) today is shaping economies, politics, and individual lives, with risks that may be more severe than those previously encountered (Markus Anderljung & Wolf, 2023; Milmo, 2025). These risks span from the amplification of individual self-harm, such as suicidal ideation among adolescents through AI chatbots (Klein, 2025), to geo-political implications of generating harmful bioweapons using AI models (Gertrude Hattoh & Akogo, 2025), and the broader societal impact of misaligned agentic AI systems (Hadshar, 2025). Prior theoretical and empirical work suggests that aligning capable AI systems with human values presents deep and unresolved challenges (Evans et al., 2024). Recently, Joshua Engels & Tegmark (2025) theoretically proved that aligning an Artificial Superintelligence (ASI) to societal values, even by using a chain of slightly inferior AI controlled by humans, is infeasible. At the same time, closed-form AI development is continuing to accelerate,

with increasing blind spots on alignment to societal values, raising the likelihood that wide-scale risks materialize before robust safeguards are established.

Historically, several beneficial innovations inadvertently produced risks that were mitigated through the creation of effective oversight and regulatory mechanisms. Industrialization and automobiles contributed to depletion of the Ozone layer, prompting international regulatory responses such as the Montreal Protocol. Advances in medicine introduced new possibilities for individualized harms, leading to institutions like the FDA to ensure safety and accountability. The current era faces a comparable inflection point as the risks of Artificial Intelligence (Chow & Perrigo, 2025; Future of Life Institute, 2023), exceed those of historical innovations, requiring societal oversight.

However, the speed of AI developments has outpaced the existing regulatory and oversight development processes. While technological changes are known to advance faster than legal and ethical governance structures can adapt, creating the well-studied 'pacing problem' (Marchant et al., 2011), AI developments have recently accelerated at an unprecedented rate. This rapid growth requires correspondingly rapid mechanisms to anticipate, mitigate, and respond to the unintended consequences of increasingly powerful AI systems at a global scale.

At the forefront, **The European Union**'s AI Act has been widely praised as a landmark policy establishing formal obligations for high-risk AI systems. While earlier drafts of the Code of Practice relied heavily on provider self-assessment, more recent versions have introduced elements of external oversight and third-party evaluation. Nonetheless, scholars continue to note that the absence of individual redress mechanisms and the deferral of key enforcement provisions to future technical standards may limit the Act's overall regulatory force (Ruschmeier, 2023; Hartmann et al., 2024; Villasenor, 2024). Furthermore, recently adopted Digital Omnibus grants the AI Office more oversight. However, studies note that the absence of redress mechanisms remains a limitation (Lazaro Cabrera & Maier, 2026; Veale & Zuiderveen Borgesius, 2021).

Across the **Asia-Pacific** region, most jurisdictions, including Japan, India, and Australia, have issued principle-based guidelines emphasizing fairness and transparency. However, these frameworks remain largely voluntary and lack binding enforcement. South Korea and Taiwan are still in the process of drafting comprehensive national legislation (Luo & Grelier, 2024). China, however, stands as a notable exception, having enacted enforceable measures such as the 2023 Interim Measures for GenAI Services, which mandate licensing, security reviews, and real-name user registration.

In contrast, the **United States**, despite being the world's leading developer of frontier AI models, has yet to pass comprehensive federal AI safety legislation. Proposals including the AI Bill of Rights and bills to establish a federal AI Safety review office, have lapsed (U.S. Congress, 2024). This, however, does not imply congressional inaction on AI regulation. On July 4, 2025, H.R.1 was enacted. Its original version included a provision imposing a 10-year moratorium on state and local regulation of AI models; this provision was removed in the Senate by a 99–1 vote (U.S. Congress, 2025). The removal signaled a clear willingness to preserve regulatory space for both state and federal oversight of AI.

Beyond formal legal mechanisms, the federal government has pursued an executive policy-based approach. It announced the "Winning the Race: America's AI Action Plan" (The White House, 2025a; CBS, 2025), which, among other goals, encourages the development of open-source models to promote innovation and transparency.

In December 2025, U.S. federal government further issued an executive order preventing AI regulation by states to ensure a national policy framework (The White House, 2025b).

Within the United States, **state-level** initiatives have started to emerge. Washington State convened an AI Task Force to advise legislators on AI Policy (Washington State Attorney General). The state is pursuing AI Safety bills such as HB1170 and HB2225 (Washington State Legislature, 2026; Washington State Standard, 2026; Fahmy, 2026; Sanford, 2026), to safeguard AI chatbot usage by minors and deepfake generated content, while California pursued bills such as SB 1047 and AB 2013 to mandate safety protocols and transparency requirements (California State Legislature, 2024). In parallel, California courts have begun shaping legal precedent in cases such as *Kadrey v. Meta*, which suggested that developers may face regulatory scrutiny when AI systems disrupt economic markets (United States District Court, Northern District of California, 2025). During the 2024 election cycle, more than 151 state bills targeted deepfakes and deceptive media, demonstrating that when risks are perceived as acute, U.S. states can act with notable speed and breadth (Stanford HAI, 2025).

While technical AI safety research is advancing, including evaluation benchmarks such as HELM (Liang et al., 2023), FLI AI Safety Index (Future of Life Institute, 2025), DecodingTrust (Wang et al., 2023), mechanistic interpretability methods such as Anthropic's 2025 circuit tracing (Ameisen et al., 2025) and structured red-teaming protocols from NIST (NIST, 2024), these remain voluntary practices rather than binding regulatory requirements. These tools can additionally supplement regulatory frameworks.

Across these diverse and limited global efforts, the unprecedented speed of AI deployments has consistently outpaced existing governance approaches.

This study contributes six key advances in support of our position: (1) a quantitative comparison between AI legislation and major AI developments demonstrating the gap in pace; (2) a comprehensive taxonomy of proposed and enacted AI policy subfields; (3) a novel dataset for studying structural and taxonomy factors of AI legislations; (4) stage-specific statistical evidence that public-interest policy sub-fields predict initial legislative engagement, while structural factors including the number of bill sponsors and chamber of origin predict failure at the committee-to-floor transition; (5) economic and developmental recommendations aimed at reducing asymmetries between policymakers and AI developers, to increase participation from all stakeholders; and (6) policy recommendations grounded in streamlined enactment through AI committees, planned adaptation, preemptive enactment, and independent AI oversight.

## 2. Analysis of U.S. Congressional Bills

To analyse the gap between AI development vs regulation and its underlying causes, we conduct a comprehensive technical analysis of all the AI related bills proposed in the U.S. Congress from 2017 to 2025. We curate the first comprehensive dataset of 150 U.S. Congressional bills, integrating major LLM releases, a taxonomy of AI subfields in bills, and legislative endpoints. For the methodology of the technical analysis and detailed sources for data curation, see Appendix A. The dataset, source code, and scripts used in this analysis are available at `www.github.com/MansurAKhan/Bridge-the-AI-Gap`.

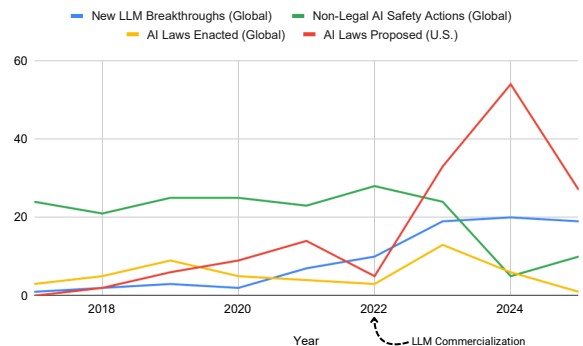

*Figure 1.* Trajectory of LLM Breakthroughs vs. Regulation for AI Safety

### 2.1. Gap in Proposed vs. Enacted Legislation

Figure 1 demonstrates temporal acceleration asymmetry: frontier LLM releases have accelerated substantially since the release of ChatGPT in 2022, while binding enactments have grown modestly, illustrating a widening governance gap. Counts reflect discrete policy events, not jurisdictional reach. It also shows that since LLM commercialization,

more than 55 new large language models have been released.

In 2023, 19 models were introduced. Another 20 during 2024, and 19 by August 2025, including GPT-5, Deep Research, and the ChatGPT Agent. However, over the same time period, only a small number of AI related laws were enacted. In 2023, 13 policies were enacted. This number declined to 6 in 2024. In 2025, only a single policy had been enacted.

The data clearly shows model development following a near-exponential growth pattern, whereas legislative enactment remains sublinear. Two additional observations emerge from the analysis. First, in 2024, non-legal AI safety actions declined by more than 50% compared to prior years, while new AI bill proposals in the United States doubled over the same period. This divergence may suggest that policymakers shifted toward legislative activity rather than symbolic gestures, although election-year dynamics may also partially explain the increase. Second, despite the rise in proposals (from 28 in 2023 to 59 in 2024), legislative outcomes remained limited (from 13 in 2023 to 6 in 2024). This further reinforces the claim that the primary gap lies not in proposing laws, but in enacting them.

Finally, of the 150 AI related bills proposed in the U.S. Congress, only three were enacted. None of these laws explicitly focuses on ensuring AI safety, highlighting the persistence of a regulatory gap despite the existence of AI related legislation. This gap is examined further through a subfield-based categorization of bills in the following subsection. A full breakdown of the data presented in Figure 1 is provided in Appendix Table 5.

### 2.2. Sub-fields of Proposed Legislation in the U.S.

While AI related laws are actively proposed and debated in the United States, the exceptionally low rate of enactment remains evident. To better understand this disparity, we next examine the technical sub-fields of these proposals. Figure 2 visualizes the distribution of sub-fields across AI related policy proposals introduced in the U.S. Congress.

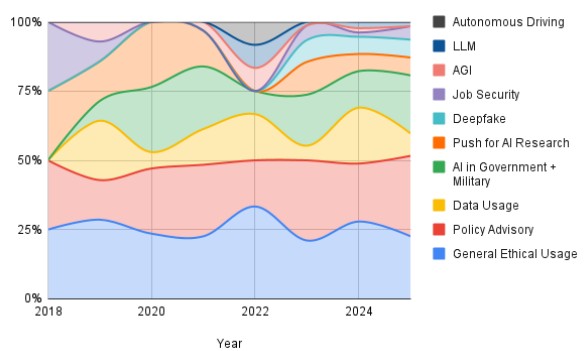

*Figure 2.* Sub-fields of Proposed Legislation in the U.S

Figure 2 reveals that since 2019, proposals concentrating on Data Usage, Policy Advisory, and General Ethical Usage have constituted the majority of introduced bills. In 2022, these three sub-fields together accounted for 66.7% of all proposed legislation, indicating that a substantial portion of congressional activity explicitly engaged with AI safety related concerns.

Shifts in sub-field emphasis further illuminate how legislative priorities have evolved over time. In 2022, the only proposals explicitly related to Large Language Models and Autonomous Driving were introduced. That same year, bills addressing Artificial General Intelligence, Large Language Models, and Autonomous Driving collectively reached an all time high of 24.9%. In contrast, proposals supporting AI research declined sharply to 0% in 2022. This marked a significant departure from earlier years, as approximately 12.9% of bills supported AI research in 2021, and 23.5% did so in 2020. Since this decline, AI research has not regained its earlier prominence, with its highest share since 2022 reaching only 11.8%.

Another notable shift emerged around deepfakes. Prior to 2023, no proposed legislation explicitly addressed this issue. By 2025, however, deepfake related provisions were included in H.R. 1 and were ultimately enacted. More broadly, before 2022, sub-fields oriented toward AI development, including Push for AI Research and AI in Government and Military, comprised a substantial portion of proposed bills. After 2022, both categories declined, while lawmakers increasingly introduced safety oriented legislation. Taken together, these trends suggest that 2022 marked a turning point, with legislative priorities shifting away from AI development and toward the establishment of safeguards.

Among all proposed sub-fields, only a small subset has resulted in multiple enacted laws. These include AI in Government and Military, Policy Advisory, and Push for AI Research. By contrast, General Ethical Usage and Deepfake legislation each resulted in only a single enacted bill. Notably, the General Ethical Usage law focuses narrowly on ethical considerations within military contexts, offering limited safeguards for broader civilian AI applications. A detailed breakdown of the data presented in Figure 2 is provided in Appendix Table 6.

## 2.3. Reasons for Failed Legislation

To understand why AI safety oriented bills are frequently proposed yet rarely enacted, we next examine the procedural paths these bills follow and the structural factors that prevent them from advancing beyond early legislative stages. We begin by analyzing the reasons for legislative failure. To quantify congressional effectiveness in advancing AI policy, we define action rate as the average action rate over the years to capture legislative progress:

$$\text{Action Rate} = \frac{1}{N} \sum_{y=1}^{N} \frac{\text{Passed Bills}_y + \text{Failed Bills}_y}{\text{Total Proposed Bills}_y} \quad (1)$$

where $N$ is the number of years.

Figure 3 shows that 89 of the 150 proposed bills stalled in committee after only initial action. Only four bills reached a terminal outcome, either passing or formally failing. Given that AI policy represents a relatively new and rapidly evolving domain, one might expect a higher level of legislative throughput. Instead, the observed action rate is only 4.23%, slightly below the 6.25% congressional average but far more volatile (SD 6.32% vs 1.92%), pointing to sporadic rather than steady AI legislating. A detailed breakdown of the data presented in Figure 3 is provided in Appendix Table 7 and the 95% CI is reported in Appendix Table 9.

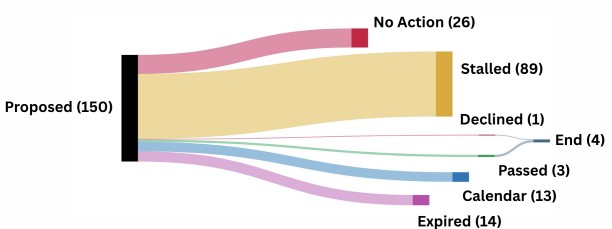

*Figure 3.* Path of Bills by Volume to Visualize the Last Stages of Each Bill

In 2023 and 2024, the years with the highest number of proposed AI related bills, 33 and 54, respectively, no bill was enacted. By contrast, in 2021, 2022, and 2025, when fewer proposals were introduced, 14, 5, and 27, respectively, a bill was enacted in each year. This pattern suggests that in AI governance, a smaller number of focused bills may have a higher likelihood of enactment than a large volume of symbolic proposals. In practice, this implies that legislative quality, including coalition building, clarity, and scope, matters more than quantity in advancing AI safety policy.

On average, 59.3% of proposed AI related bills stalled at the committee or subcommittee stage, as shown in Figure 3. These bills were most often referred to committees overseeing technology, ethics, or economic policy. In practice, AI regulation proposals are frequently sent to subcommittees, where they are placed on a calendar and receive no further action. In legislative procedure, this outcome is commonly referred to as pigeonholing. One contributing factor is that within committees and subcommittees, "if the leadership decides the bill does not fit within its overall agenda, a decision not to act will 'kill' the bill just as effectively as a vote against it" (ProQuest, 2025). These dynamics point to a structural limitation of existing legislative pathways and motivate the need to examine additional factors that contribute to bill stagnation.

## 2.4. Further Factors Contributing to Stalling

*Table 2.* Hurdle Model Stage 1 Coefficients with AI Topic Sub-fields Bolded

| Feature | Coefficient | P-value | FDR Q | 95% CI | \|Coefficient\| |
|---|---|---|---|---|---|
| **Deepfake** | 1.5536 | *1.21e-6 | *2.91e-5 | [0.9126, 2.1491] | 1.5536 |
| **Job Security** | 1.2298 | *0.0005 | *0.0056 | [0.5397, 1.9126] | 1.2298 |
| **Push for AI Research** | -1.1933 | 0.0739 | 0.3548 | [-2.3997, 0.2376] | 1.1933 |
| **Data Usage** | 0.4126 | 0.4728 | 0.7091 | [-0.6239, 1.7716] | 0.4126 |
| **General Ethical Usage** | -0.3736 | 0.5101 | 0.7201 | [-1.3696, 0.8330] | 0.3736 |
| **Policy Advisory** | 0.1283 | 0.8239 | 0.9416 | [-1.0553, 1.1819] | 0.1283 |
| **Advanced AI** | -0.0395 | 0.9616 | 0.9616 | [-1.4988, 1.3367] | 0.0395 |
| **AI in Government + Military** | -0.0285 | 0.9603 | 0.9616 | [-1.1610, 1.1313] | 0.0285 |
| Sponsor_Party_Binary | -0.9085 | 0.1838 | 0.4448 | [-2.2694, 0.3403] | 0.9085 |
| Num_Sponsors | -0.3570 | 0.3890 | 0.6224 | [-1.0237, 0.4881] | 0.3570 |
| Bipartisan | -0.2228 | 0.7249 | 0.8699 | [-1.5736, 0.9662] | 0.2228 |
| Chamber_Binary | 0.0901 | 0.8864 | 0.9616 | [-1.2436, 1.2360] | 0.0901 |

*Notes:* $^*$p<0.05; FDR Q<0.05.

*Table 3.* Hurdle Model Stage 2 Coefficients with AI Topic Sub-fields Bolded

| Feature | Coefficient | P-value | FDR Q | 95% CI | \|Coefficient\| |
|---|---|---|---|---|---|
| **Advanced AI** | 0.9790 | 0.1944 | 0.4448 | [-0.5817, 2.3475] | 0.9790 |
| **Deepfake** | -0.7584 | 0.1576 | 0.4448 | [-1.6719, 0.3167] | 0.7584 |
| **Push for AI Research** | -0.7256 | 0.1589 | 0.4448 | [-1.7578, 0.3107] | 0.7256 |
| **General Ethical Usage** | -0.6630 | 0.1955 | 0.4448 | [-1.7098, 0.3298] | 0.6630 |
| **Policy Advisory** | -0.5080 | 0.3308 | 0.6224 | [-1.4953, 0.4767] | 0.5080 |
| **Job Security** | 0.5049 | 0.3552 | 0.6224 | [-0.6464, 1.4481] | 0.5049 |
| **Data Usage** | -0.4738 | 0.3658 | 0.6224 | [-1.5419, 0.5096] | 0.4738 |
| **AI in Government + Military** | 0.1901 | 0.7010 | 0.8699 | [-0.7268, 1.1887] | 0.1901 |
| Chamber_Binary | -1.0693 | *0.0416 | 0.2499 | [-2.1392, -0.0385] | 1.0693 |
| Num_Sponsors | -0.9765 | *0.0102 | 0.0815 | [-1.7897, -0.3114] | 0.9765 |
| Bipartisan | 0.6650 | 0.2039 | 0.4448 | [-0.3826, 1.6924] | 0.6650 |
| Sponsor_Party_Binary | 0.2415 | 0.5997 | 0.7996 | [-0.6415, 1.1567] | 0.2415 |

*Notes:* $^*$p<0.05; FDR Q<0.05.

We next examine the factors that most strongly contribute to legislative stalling across distinct procedural stages, with the aim of identifying mechanisms that can more effectively address the enactment gap. To preserve the statistical power of the dataset and disentangle the predictors of initial legislative engagement from those of late-stage advancement, we implement a Sequential Binary Hurdle Model (also known as a Two-Stage Choice Model; (Krutz, 2005; Cragg, 1971)). This approach models two distinct, conditional thresholds that define a bill's trajectory: Hurdle 1, the Engagement Gate, modelling the transition from inaction to committee engagement ($n = 115$); and Hurdle 2, the Advancement Gate, modelling the transition from committee to stalled or advanced/resolved ($n = 102$).

Tables 2 and 3 present the bootstrapped mean coefficients, p-values, FDR Q-values, and 95% confidence intervals for each stage. Sub-fields in bold represent AI policy topic variables; control variables are separated by a horizontal rule. This granular analysis reveals a stage-specific pattern.

At Hurdle 1, policy sub-fields exhibit the strongest predic-

tive power. Bills addressing Deepfakes (coefficient = 1.5536, $p = 1.21 \times 10^{-6}$, FDR $Q = 2.91 \times 10^{-5}$) and Job Security (coefficient = 1.2298, $p = 0.0005$, FDR $Q = 0.0056$) are substantially more likely to clear the engagement gate, with both results remaining significant after FDR correction. This suggests that bills addressing salient, public-interest concerns are more likely to attract initial legislative attention. The remaining sub-fields do not reach statistical significance at this stage, indicating that this analysis cannot confirm a systematic relationship between other policy domains and the probability of initial engagement. Structural and political control variables similarly show no statistically significant association with committee engagement, suggesting that procedural and coalition factors do not play a primary role at this early threshold.

At Hurdle 2, the pattern reverses. Policy sub-fields lose their predictive power entirely, with no topic variable approaching statistical significance. Instead, structural predictors emerge. The number of bill sponsors (coefficient = $-0.9765$, $p = 0.0102$) and chamber of origin (coefficient = $-1.0693$, $p = 0.0416$) are both significant at the conventional threshold, though neither survives FDR correction ($Q = 0.0815$ and $Q = 0.2499$, respectively). These associations are correlational and may be influenced by omitted variables such as committee assignment, bill salience, and timing, and should not be interpreted as causal. Nevertheless, political science literature offers several explanations for the direction and magnitude of these coefficients. Sotoudeh et al. (2024) show that broader cosponsorship by several moderately influential representatives instead of a few influential representatives can reduce the likelihood of advancement through committee stages. Kessler & Krehbiel (1996) show that cosponsorship can serve as a strategic signal without guaranteeing passage. This is in line with Riker (1962) minimum winning coalition theory, which argues that rational actors prefer coalitions that are just large enough to win, as larger coalitions dilute payoffs. For AI Bills in the US Congress, a high sponsor count likely proxies for a signaling statement or slowdown of the legislative process, instead of optimizing for bill passage.

Taken together, these findings sharpen and substantially revise our earlier conclusion. The procedural bottleneck is not uniformly distributed across the legislative pipeline: public-interest topics correlate with gaining early traction at the engagement stage, but do not predict advancement through later procedural barriers. Conversely, structural and political factors align specifically with the committee-to-floor transition. Rather than policy content or procedural dynamics alone, it is their stage-specific interaction that determines whether AI safety legislation advances.

# 3. Why Does the Gap Persist?

The hurdle model reveals a stage-specific pattern: at Hurdle 1, bills addressing Deepfakes and Job Security are significantly more likely to clear the engagement gate, while structural and political control variables show no significant association at this early threshold. At Hurdle 2, policy subfields lose predictive power entirely, and structural factors, specifically the number of bill sponsors and chamber of origin, emerge as the dominant correlates of advancement failure, though neither survives FDR correction. However, quantitative findings alone do not fully explain the observed enactment gap. Broader procedural and informational constraints as well as political and economic incentives provide a more comprehensive account of why AI policy has struggled to keep pace with technological development. To complement the statistical analysis of Table 2 and Table 3, we next examine these constraints and incentives in detail.

## 3.1. Fast-paced Development and Economic Incentives

AI systems are evolving rapidly and unpredictably, which makes it difficult for regulatory frameworks to anticipate emerging capabilities in advance. As a result, there is lack of oversight during the phase of accelerated innovation (Pouya Kashefi & Mirsaraei, 2024).

Economic incentives have significantly compressed AI development timelines. Empirical and conceptual research shows that AI research and deployment operate within a highly competitive environment in which firms and research groups race to achieve technical breakthroughs, since early leadership yields disproportionate economic and strategic advantages (Hartmann et al., 2024). Competitive pressure also encourages organizations to adopt and scale AI systems quickly in order to maintain market position, reinforcing incentives that prioritize speed over safety considerations (Wong & Ngai, 2025). Studies of AI race demonstrate that competition among firms and states shortens development cycles and discourages cautious or alignment focused approaches, even when such approaches may reduce systemic risk (Gruetzemacher et al., 2025). Analysis of industry firms reveals market pressures incentivizing rapid capability release ahead of mature governance or safety evaluation (Zoë B. Cullen, 2025).

This persistent mismatch between the pace of innovation and the speed of response from institutions results in safety and alignment considerations being deprioritized relative to deployment and competition. Addressing this imbalance requires economic and regulatory incentives that reward alignment oriented development (Reuel & Undheim, 2024; West, 2023).

## 3.2. Institutional and Procedural Constraints

Legislatures are inherently designed to preserve the status quo unless change becomes unavoidable, a tendency often described as legislative inertia (Cohen, 2021). This characteristic makes the design of new legislation, or the adaptation of existing laws, particularly challenging when addressing novel technological risks. Historical research shows that updating legal frameworks in response to technological change is bureaucratically costly and slow (Kingdon, 1984), even though such processes can produce highly effective and adaptable policy regimes when deliberately structured through approaches such as planned adaptation (McCray et al., 2009). Legislative inertia also helps explain why, even when bills are introduced with strong intentions, committees can pigeonhole proposals, effectively terminating them without formal opposition (ProQuest, 2025; Wiseman, 2023; Marchant et al., 2011). This pattern is consistent with our empirical finding that approximately 59% of AI related bills stall at the committee stage, as shown in Figure 3.

Bills are routinely referred to committees or subcommittees, where agenda setting authorities can slow or deprioritize proposals deemed to be of lower urgency, preventing many bills from reaching floor consideration (ProQuest, 2025; GovTrack, 2004; Wiseman, 2023). In the context of AI legislation, committee fragmentation further compounds this challenge. Oversight responsibilities often span multiple committees, including Commerce, Judiciary, and Defense, which increases coordination burden and delays consensus on both the scope and substantive content of proposed legislation.

## 3.3. Information and Expertise Gaps

Another major contributor to regulatory delay can be understood through the economic concept of information asymmetry. In this context, the asymmetry arises from disparities in technical expertise between lawmakers and AI developers. Rapid advances in AI research generate highly specialized knowledge that is often difficult for legislators to access or evaluate directly (Wei et al., 2024; Pouya Kashefi & Mirsaraei, 2024; Reuel & Undheim, 2024; Carey, 2025). As a result, congressional committees rely heavily on staff, external experts, and industry stakeholders to assess the societal implications of AI development. These intermediaries may hold competing incentives or priorities, which can prolong deliberation and complicate consensus building (Costa et al., 2015; Albino et al., 2013; Elizabeth Rybicki, 2023; Ugarte, 2026; Wei et al., 2024).

The principal agent structure of lawmaking further exacerbates these delays. Elected representatives act as agents for voters but must navigate complex technical information supplied by factors such as developers, firms, investors, and researchers, each with distinct interests (Costa et al.,

2015; Albino et al., 2013; Elizabeth Rybicki, 2023). This dynamic increases informational complexity and slows the enactment of policy (Wei et al., 2024; Pouya Kashefi & Mirsaraei, 2024). Consequently, committee debates often evolve into extended negotiations over scope, language, and enforceability, rather than straightforward approval of broadly supported safeguards (Carey, 2025; Ugarte, 2026; ProQuest, 2025).

### 3.4. Political Incentives and Collective Action

Political incentives and collective action problems further impede legislative progress. Industry coalitions are often highly organized and capable of shaping or slowing legislation in ways that align with their strategic interests. Evidence from recent California AI legislation and the European Union AI Act omnibus package illustrates how broad technology alliances have resisted stricter regulatory provisions, instead advocating for simplified or narrower liability frameworks (Minkin, 2025). Because general purpose AI technologies remain unfamiliar to many lawmakers, committees must carefully assess potential harms, ethical implications, and long term societal impacts before drafting enforceable rules (Carey, 2025) as AI policy often lacks a clear constituent narrative (West, 2023). This uncertainty increases reliance on industry coalitions for information and guidance.

Short electoral cycles also incentivize legislators to prioritize issues that yield rapid and visible benefits, often relegating perceived long term and controversial risks, such as AI, to a secondary status (Kingdon, 1984; OECD, 2025). At the same time, partisan disagreements over innovation, privacy, and national security amplify procedural delays, as committees must reconcile competing ideological positions before advancing bills to the floor (Wiseman, 2023).

## 4. Alternative Views

Debates around AI governance frequently raise concerns related to geopolitical competition, economic incentives, institutional feasibility, and the timing of alignment and oversight. While these perspectives reflect legitimate anxieties about innovation, competitiveness, and security, we argue that many rely on assumptions that, when examined reveal clear directions for simultaneous technological progress and oversight. Across these views we find a common theme that prioritizing speed, fragmentation, or post hoc intervention ultimately amplifies systemic risk rather than mitigating it.

**The Global Conflict Position.** Some policymakers and governance advocates argue that ensuring strategic advantage in AI development requires prioritizing speed over institutional oversight. From this perspective, slowing innovation to accommodate alignment or regulatory processes risks

allowing geopolitical adversaries with incompatible values to gain control over advanced AI systems. As a result, institutional safeguards and alignment efforts are treated as secondary concerns.

*Our response.* While this position is grounded in the ethical concern of preventing misuse by malicious factors, we argue that continuous acceleration driven by geopolitical competition is ultimately counterproductive. Persistent race dynamics amplify systemic risks and increase the likelihood of irreversible harm. Rather than equating speed with security, we argue that long-term stability depends on building institutional trust and enforceable ethical compliance into AI systems. By treating alignment and governance as sources of strategic strength rather than constraints, control over AI can be both broadly distributed and anchored in shared human values.

**To What Extent Should AI Be Regulated?** Regulation should be stratified by capability and risk. Consumer applications with individual risks should require transparency and consent frameworks such as those in WA state legislation HB2225. Broader-risk systems, such as those capable of generating bioweapons, should require a mandatory predeployment safety evaluation. A threshold-based framework should trigger oversight at defined capability benchmarks (dangerous capabilities, deployment scale), to concentrate regulation on riskier systems.

**The Global Economic Incentives Argument.** Some contend that stringent national regulation will slow AI development, allowing other nations to capture economic benefits and technological leadership. Under this view, unilateral regulation risks placing early movers at a disadvantage in global markets.

*Our response.* We argue that coordinated standards and accountability frameworks across major economies, particularly the United States, the European Union, and Asia-Pacific countries, can mitigate this risk. Alignment across these regions reduces regulatory arbitrage while preserving innovation incentives, enabling AI safety governance without ceding economic competitiveness. For example, the Montreal Protocol, 197 nations aligned despite initial resistance from major companies such as DuPont, which actively opposed regulations to protect profits. Recently, highly competitive firms started to align on safety standards. The EU GPAI Code of Practice (2025) was signed by Amazon, Google, Microsoft, OpenAI, and Anthropic, all direct competitors who nonetheless agreed to shared oversight provisions (European Commission, 2025). These indicate that international alignment in AI is also feasible. Additionally, we advocate minimum standards (analogous to the Basel Accords in banking).

**The Local Economic Incentives Argument.** At the firm

level, companies argue that delaying AI development for safety or regulatory compliance allows competitors to move faster and capture market share.

*Our response.* Regulation implemented at the state or national level preserves competitive parity within local markets. When all firms operate under the same rules, no individual actor faces a structural disadvantage. In this context, regulation functions as a coordination mechanism that protects competition while reducing systemic risk.

**The Afterthought Argument.** Some critics maintain that alignment with human values need not be embedded from the outset, as developers can retrofit safety mechanisms after deployment. They often cite firm-led safety evaluations, such as OpenAI's assessments of GPT-5, as evidence that post hoc alignment is sufficient (OpenAI, 2025).

*Our response.* We argue that treating alignment as an afterthought is both economically inefficient and risk-amplifying. Designing systems with built-in alignment reduces the need for repeated corrective interventions and lowers downstream safety costs. Moreover, despite reported safety efforts, empirical evidence indicates rising incidents of LLM-mediated self-harm in 2025 and increased reports of models exhibiting self-preserving or harmful behaviors in simulated environments (Klein, 2025; Meinke et al., 2024; Evans et al., 2024). Embedding alignment into system design, rather than retrofitting it under pressure, offers a more robust path to risk reduction.

**The Implementation Concern.** Even among those who agree that regulation should keep pace with technological innovation, skepticism remains about whether this goal is achievable. This concern often stems from economic and political incentives that slow regulatory action, the rigidity of institutional design, and uncertainty about stakeholder willingness to implement adaptive governance mechanisms. As a result, critics question whether the gap between AI development and regulation can realistically be bridged.

*Our response.* Drawing on historical precedent, we argue that bridging this gap is not only possible but has been achieved repeatedly in other high-risk technological domains. The rise of carbon emissions, the health risks posed by unverified medical treatments, and the development of nuclear weapons each introduced unprecedented societal risks. In response, regulatory institutions and governance frameworks were eventually established to manage these challenges effectively (Lee, 2007; Webster, 2009; Judge et al., 2021; OECD, 2021).

## 5. Call to Action

Bridging the gap between AI development and effective governance requires coordinated action across technical, institutional domains. We call for the following.

### 5.1. For Developers

Developers design and deploy AI systems and therefore bear direct responsibility for embedding safety, robustness, and alignment throughout the development lifecycle. We call on developers to:

1. **Built-in Alignment:** Integrate alignment evaluation as a first-order engineering gate, through safety cases, red-teaming, and pre-deployment review, rather than treating it as a post-deployment correction.

2. **Scientist-Oriented AI Paradigms:** Adopt development paradigms similar to "scientist AI," in which systems are optimized to understand, explain, and reason carefully, rather than to maximize task completion or persuasion. Removing implicit intent from model objectives can reduce harmful or misleading outputs.

3. **Engage with Legislators:** Proactively engage with policymakers and oversight institutions to inform AI regulation grounded in technical realities and societal risk considerations, enabling evidence-based and implementable policy design.

### 5.2. For Policymakers

Policymakers design the institutions, regulations, and legislation that serve the public interest. We call on policymakers to:

1. **Establish Dedicated AI Policy Committees:** Our analysis shows that committee pigeonholing is a dominant cause of legislative stalling. Legislatures should establish standing committees or subcommittees dedicated exclusively to AI policy and ethics in both the House and Senate. Concentrating expertise within a single accountable body would reduce procedural bottlenecks and accelerate informed oversight.

2. **Implement Preemptive Enactment Models:** To reduce lag between technological advancement and governance, policymakers should adopt preemptive regulatory frameworks that activate oversight mechanisms once defined technical thresholds are reached. Models exceeding specified computational scale or trained on sensitive data should automatically trigger audits, reporting requirements, or deployment restrictions.

3. **Introduce Sunset Clauses:** Sunset clauses (provisions that cause legislation to automatically expire unless actively renewed) are well-suited to AI governance for three reasons. First, rapid advancements in AI can render regulations obsolete; the expiration due to

sunset clauses forces regular reassessment by legislators. Second, sunset clauses lower the initial political cost of enactment by framing new regulation as temporary and reversible, which can reduce opposition from both industry and cautious legislators. Third, they create built-in feedback loops where each renewal cycle generates an intervention for updating based on new technical evidence and stakeholder input.

For AI specifically, we propose the following design: AI safety legislation should include a 5-year sunset with review at 3-year mark. The review would be conducted by the independent AI safety agency we propose to avoid the pigeonholing dynamics we document. Traditionally, sunset clauses are avoided to ensure that legislation isn't done without future foresight; however, this can be addressed via review mechanisms.

4. **Create Independent AI Safety Agencies:** Independent agencies, analogous to the FDA or FAA, could centralize technical expertise, conduct compliance audits, and enforce AI safety standards. Such institutions would reduce reliance on ad hoc legislative hearings and enable faster responses to emerging risks.

5. **Directional Propagation of Terminal outcomes**: global alignment should set the baseline, regional regulation dictate the binding framework, and national execution to determine the real-world safety outcome. Because a failure at any level creates a 'weak link' that can compromise the entire envelope, success gates should be cleared to prevent regulatory leakage:

   - International Alignment Gate: Formalization of shared safety principles on catastrophic risks, to ensure global interoperability and prevent cross-border gaps. Measured through adoption by major global regions.
   - Regional Framework Gate: Success is the entry into force of a binding regulation and centralized oversight body, (e.g. the EU AI Act and EU AI Office). This creates a legal floor for the region.
   - National Operationalization Gate: Success is the activation of domestic enforcement infrastructure across all states. This requires each country to enact national legislation for example to designate competent authorities and set penalty structures, such as Spain's AESIA (EU, 2025).

**The 'Weak Link' and Feedback Loops:** For example, in the EU context, the AI Act is a directly applicable Regulation, yet its efficacy depends on national operational success. If a subset of countries fails to operationalize their local authorities, it creates regulatory leakage: non-compliant products may enter through a 'weak' member state and then move across the 27-state market, undermining the regional safety floor. We view these localized failures as both: risks as well as essential feedback loops. They highlight friction points which could inform the future refinement of the regional framework. Ultimately, true success is not merely the passage of law, but the unified clearance of the Operationalization Gate across the entire envelope.

### 5.3. For Economists

Economists analyze incentives, market structure, and long-term economic impacts of policy choices. We call on economists to:

1. **Quantify Innovation and Safety Trade-offs:** Evaluate how different regulatory approaches affect startup growth, research investment, and international competitiveness, helping policymakers calibrate rules that minimize economic disruption while ensuring safety remains the priority.

2. **Design Market-Based Safety Incentives:** Develop insurance mechanisms, liability regimes, and certification systems that internalize AI-related risks and create financial incentives for safer system deployment.

3. **Monitor Regulatory Capture Risks:** Analyze how concentrated lobbying and market power influence AI legislation, and advocate for transparency and counterbalancing mechanisms that limit disproportionate influence by large firms.

## 6. Conclusion

This paper demonstrates that the widening gap between rapid AI development and legislative response is driven by procedural and informational constraints as well as political and economic incentives, that impact development and governance institutions.

Our empirical analysis shows that public-interest topics are predictive early committee engagement, while structural factors such as sponsor count and chamber of origin are associated with late-stage stalling (though structural feautures lose significance post-FDR correction) reinforcing theories of legislative inertia. Fast-paced technological change is increasing the risks of AI by outpacing the response mechanisms of traditional governance institutions.

Together, these forces create a sustained mismatch between the speed of innovation and the ability of governance structures to respond effectively. Addressing this gap requires dedicated AI committees that reduce procedural bottlenecks such as pigeonholing, and adaptive legislation through preemptive enactment and sunset clauses, informed by all stakeholders, without undermining democratic accountability. Without such changes, AI policy is likely to remain reactive after risks materialize.

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

# Appendix

## A. Methodology for Technical Analysis

This paper employs a mixed-methods approach combining quantitative legislative analysis with qualitative policy evaluation to examine structural bottlenecks in U.S. AI safety regulation, while situating these findings within a broader international context. The study focuses not only on the volume of proposed and enacted legislation, but also on the procedural mechanisms that inhibit legislative progress, particularly within the United States. All code, scripts, results, and datasets used in this analysis are publicly available at `www.github.com/MansurAKhan/Bridge-the-AI-Gap`.

### A.1. Data Sources and Compilation

To construct a systematic comparison between AI development and AI regulation, this paper aggregates data from several verified public and governmental sources. Unlike prior trackers, we assemble the first comprehensive dataset spanning 2017 through August 2025 that integrates bill sub-fields, legislative endpoints, and modeled bottleneck factors.

- **AI Breakthroughs:** Major large language model (LLM) releases from 2017 to mid-2025 were compiled using the *LLM Timeline Project*, Wikipedia's *List of Large Language Models*, and official publication announcements from leading developers such as OpenAI, DeepSeek, Mistral, DeepMind, and Anthropic.

- **U.S. Legislation:** AI-related bills proposed at the federal level were retrieved from the official U.S. Congress database (`www.congress.gov`)[1], filtered using the keywords "Artificial Intelligence" and "AI". Additional records were sourced from the National Conference of State Legislatures (NCSL) and the Brennan Center for Justice. Each bill was categorized into sub-fields. This is the first comprehensive categorization of all AI bills into sub-fields.

  The sub-fields were formulated by a human subject matter expert and are verified by a lawyer. The classification of the sub-fields was done by providing the sub-fields and the URLs of all 150 bills for GPT-4o to output any sub-fields that correspond to each bill, and provide a confidence level (Low, Medium, High). Bills with a low confidence rate were audited and corrected manually. The accuracy of the labeled data was determined by selecting 50 classifications at random and verifying the accuracy through manual auditing. The accuracy was found to be 94% (47 out of 50).

  Through comprehensive manual human annotation, each bill was assigned an endpoint and a systematic rationale for reaching this point. This is the first comprehensive analysis of endpoints of AI bills. The following is a breakdown of each ending category:

  - **Expired without action:** This label is attributed to the bills that have only one action regarding them, be it a referral to a committee or a hearing. If the bill does not step through the introduction, it is marked as "Expired without action."
  - **Stalled in Committee (House/Senate):** This label is attributed to the bills that have multiple actions regarding them, and their last action is to be referred to a Committee, which can happen in both the Senate and the House.
  - **Senate/House Calendar Inaction:** This label is attributed to the bills that have multiple actions regarding them, referred to and passed a committee, and their last action is to be placed on Calendar, which can happen in both the Senate and the House. This process can occur multiple times, but none of the failed bills have reached this stage.
  - **No Action After Introduction (note):** This label is attributed to the bills that are from the currently occurring 119th Session of Congress. In brackets, a note is provided to them to show their latest action, not a category.
  - **Declined:** This label is attributed to the bills that reached an end decision not to be enacted.
  - **Passed:** This label is attributed to the bills that reached an end decision to be enacted.
  - **Amendment Passed:** This label is attributed to the amendments of bills that reached a decision to be enacted.

- **International Policy:** Global enactment and policy activity were gathered using the European Council's AI legislation tracker, the OECD's AI Policy Observatory, and Asia-Pacific legal reports (e.g., *InsideGlobalTech*). International summits, declarations, and non-legislative initiatives were cross-verified through news archives and official government publications.

---

[1]The HAI Index Report 2025 contains a dataset covering proposed bills, amendments, and similar legislative action in the United States, similar to the "US AI Laws Proposed" dataset count (view Appendix Table 5). The methods applied for analysis in this paper are only possible on bills; thus, we created a separate analytical dataset of the bills, available at `www.github.com/MansurAKhan/Bridge-the-AI-Gap`.

## A.2. Policy Classification Criteria

Each policy item was classified into one of the following categories:

- **Proposed Legislation:** Any AI-related bill formally introduced into a national or supranational legislature.

- **Enacted Legislation:** Policies that successfully passed both legislative branches and came into legal effect.

- **Non-Legal Action:** Includes summits, executive orders, AI task forces, public safety frameworks, and ethical guidelines that lack binding authority.

- **Failed Bill:** A bill that did not reach a final decision but failed in the process of getting to one. If a bill did not pass, it is a failed bill.

U.S. legislative outcomes were further coded into paths: stalled in committee, calendar inaction, declined, passed, or no action after introduction. These were used to calculate the **Action Rate**:

$$\text{Action Rate} = \frac{1}{N} \sum_{y=1}^{N} \frac{\text{Passed Bills}_y + \text{Failed Bills}_y}{\text{Total Proposed Bills}_y}$$

This metric serves as a representative for congressional engagement and legislative momentum.

## A.3. Analytical Framework

Trends were analyzed year-by-year from 2017 through July 2025. Visualizations (e.g., Sankey diagrams, bar charts) were created to represent the results through a visual medium. Policy bottlenecks were interpreted using committee records and external legislative studies, such as ProQuest Congressional Insight and the Stanford HAI AI Index Report (2025).

## A.4. Deduction of Legislative Factors Attributed to Stalling

To identify the factors that influence legislative outcomes at distinct procedural stages, the human annotated dataset was expanded using data retrieved through the Congress.gov API, accessed via https://www.congress.gov/help/using-data-offsite. This API provides machine readable legislative metadata that can be viewed, retrieved, and reused for analysis. Using these data, we constructed a feature set consisting of chamber of origin, sponsor party, bipartisan status, number of sponsors, and policy sub-field indicators, yielding a total of 12 parameters.

Of the 150 AI related bills in the dataset, 124 reached a terminal legislative endpoint, indicating that the congressional session to which they belonged expired and no further action could be taken. The remaining 26 bills belong to the ongoing 119th Congress and have not yet reached a terminal outcome. Although their current status may reflect stalling, the full set of contributing factors cannot yet be observed. Accordingly, these bills were excluded from both stages of the model.

Rather than treating legislative stalling as a single binary outcome, we implement a Sequential Binary Hurdle Model (also known as a Two-Stage Choice Model; Krutz, 2005; Cragg, 1971) to disentangle the predictors of initial legislative engagement from those of late-stage advancement. This approach models two distinct, conditional thresholds that define a bill's trajectory:

- **Hurdle 1 (The Engagement Gate):** A binary outcome modelling the transition from inaction to committee engagement ($n = 115$). Bills that received any form of committee action are coded as having cleared this hurdle.

- **Hurdle 2 (The Advancement Gate):** A binary outcome modelling the transition from stalled-in-committee to advanced or resolved ($n = 102$), conditional on having cleared Hurdle 1. Bills that progressed beyond committee are coded as having cleared this hurdle.

Each stage is modelled independently using a penalized logistic regression with ridge regularization (Le Cessie & van Houwelingen, 1992), trained and executed on Google Cloud using a virtualized Intel Xeon CPU operating at 2.20GHz. All results were analyzed and interpreted using Python based statistical libraries.

To improve model stability and interpretability, several low frequency sub-fields, including Large Language Models, Artificial General Intelligence, and Autonomous Driving, were aggregated into a broader Advanced AI category. Individually, these sub-fields contained only one or two observations, which resulted in unstable coefficients and inflated variance. Grouping them reduced volatility and produced more reliable estimates.

Both hurdle models were implemented using scikit-learn with an L2 regularization penalty. Hyperparameters were held constant across both stages as follows: regularization parameter $C$ equal to 1.0, maximum iterations set to 2000, solver specified as `lbfgs`, and class weighting set to balanced to account for outcome imbalance. Bootstrap resampling with 1000 iterations was employed for each stage, with each iteration splitting the data into 80% training and 20% testing subsets. Reported coefficients represent bootstrapped means, and p-values were derived from z-scores computed using the bootstrapped standard errors. Feature scaling was performed using StandardScaler with z-score normalization.

To guard against false positives arising from multiple hypothesis testing across the 12 features in each stage, we applied the Benjamini–Hochberg False Discovery Rate (FDR) procedure in place of a Bonferroni correction, preserving statistical power while controlling the expected proportion of false discoveries. Statistical significance was assessed at an $\alpha$ level of 0.05 for both unadjusted p-values and FDR Q-values.

# B. Further Clarity

## B.1. Sunset Clauses Suitability for AI Governance

Sunset clauses (provisions that cause legislation to automatically expire unless actively renewed) are well-suited to AI governance for three reasons. First, rapid advancements in AI can render regulations obsolete; the expiration due to sunset clauses forces regular reassessment by legislators. Second, sunset clauses lower the initial political cost of enactment by framing new regulation as temporary and reversible, which can reduce opposition from both industry and cautious legislators. Third, they create built-in feedback loops where each renewal cycle generates an intervention for updating based on new technical evidence and stakeholder input.

For AI specifically, we propose the following design: AI safety legislation should include a 5-year sunset with review at 3-year mark. The review would be conducted by the independent AI safety agency we propose to avoid the pigeonholing dynamics we document. Traditionally, sunset clauses are avoided to ensure that legislation isn't done without future foresight; however, this can be addressed via review mechanisms.

## C. Rebuttal to Alternate Views on AI Governance

| Alternate View | Rebuttal |
| --- | --- |
| **Global Conflict Position** | Speed is prioritized to prevent adversarial nations from gaining control over advanced AI systems, even if this requires weakening oversight and alignment efforts. Persistent race dynamics increase systemic risk and raise the likelihood of irreversible harm. Long-term strategic stability is better achieved through institutional trust and enforceable ethical compliance. Treating alignment and governance as sources of strategic strength, rather than constraints, enables durable control of AI aligned with shared human values. |
| **Global Economic Incentives Argument** | Strict national regulation risks slowing innovation and allowing other nations to capture economic and technological leadership. Coordinated standards across major economies, including the United States, the European Union, and Asia-Pacific countries, reduce regulatory arbitrage while preserving innovation incentives. International alignment enables AI safety governance without sacrificing economic competitiveness. |
| **Local Economic Incentives Argument** | Firms argue that safety requirements disadvantage compliant companies relative to faster moving competitors. Regulation applied uniformly at the state or national level preserves competitive parity. When all firms operate under the same rules, no individual actor faces structural disadvantage. Regulation thus functions as a coordination mechanism that protects competition while reducing systemic risk. |
| **Afterthought Argument** | Alignment can be added after deployment through safety evaluations and post hoc safeguards, making early integration unnecessary. Post hoc alignment is economically inefficient and risk amplifying. Built-in alignment reduces downstream correction costs and lowers safety risk from the outset. Empirical evidence shows rising LLM-mediated self-harm and harmful agent behavior despite reported safety efforts, indicating that retrofitting alignment is insufficient. |
| **Necessity Concern** | Existing regulatory frameworks, such as the EU AI Act, already balance innovation and oversight adequately. Current frameworks remain fragile and insufficiently adaptive. Omnibus revisions can dilute accountability, and lobbying pressures can delay or weaken enforcement. Ongoing AI risks indicate that existing regulation does not fully close the governance gap and may allow it to widen. |
| **Implementation Concern** | Even if regulation is desirable, institutional, political, and economic barriers make implementation impractical. Historical precedent demonstrates that governance can catch up to high-risk technologies, including emissions control, medical safety, and nuclear policy. Early AI regulation, such as the EU AI Act, already shows that enforceable obligations on general-purpose models are feasible. Bridging the gap is implementable within existing institutional frameworks. To What Extent Should AI Be Regulated? Regulation should be stratified by capability and risk. Consumer applications with individual risks should require transparency and consent frameworks such as those in WA state legislation HB2225. Broader-risk systems, such as those capable of generating bioweapons, should require a mandatory pre-deployment safety evaluation. A threshold-based framework should trigger oversight at defined capability benchmarks (dangerous capabilities, deployment scale), to concentrate regulation on riskier systems. |

*Table 4.* Summary of Alternate Views on AI Governance and Corresponding Rebuttals

# D. Data Tables

*Table 5.* Comparison of LLM Breakthroughs and AI Safety Activity (2017–2025)

| Year | LLM Breakthroughs | US AI Laws Proposed | AI Laws Enacted (Global) | Non-Legal AI Safety Actions |
|---|---|---|---|---|
| 2017 | 1 | 0 | 3 | 24 |
| 2018 | 2 | 2 | 5 | 21 |
| 2019 | 3 | 6 | 9 | 25 |
| 2020 | 2 | 9 | 5 | 25 |
| 2021 | 7 | 14 | 4 | 23 |
| 2022 | 10 | 5 | 3 | 28 |
| 2023 | 19 | 33 | 13 | 24 |
| 2024 | 20 | 54 | 6 | 5 |
| Jan–Aug 2025 | 19 | 27 | 1 | 10 |

*Note:* Data from 2025 is not comprehensive, as it was compiled in Aug 2025.

*Table 6.* Sub-fields of Proposed AI Legislation in the U.S. (2018–Jul 2025)

| Year | General Ethical Usage | Policy Advisory | Data Usage | AI in Gov/Military | Push for AI Research | Deepfake | Job Security | AGI | LLM | Autonomous Driving |
|---|---|---|---|---|---|---|---|---|---|---|
| 2018 | 1 | 1 | 0 | 0 | 1 | 0 | 1 | 0 | 0 | 0 |
| 2019 | 4 | 2 | 3 | 1 | 2 | 0 | 1 | 1 | 0 | 0 |
| 2020 | 4 | 4 | 1 | 4 | 4 | 0 | 0 | 0 | 0 | 0 |
| 2021 | 7 | 8 | 4 | 7 | 4 | 0 | 0 | 1 | 0 | 0 |
| 2022 | 4 | 2 | 2 | 1 | 0 | 0 | 0 | 1 | 1 | 1 |
| 2023 | 16 | 22 | 4 | 14 | 9 | 6 | 4 | 0 | 1 | 0 |
| 2024 | 36 | 27 | 26 | 17 | 8 | 8 | 2 | 2 | 3 | 0 |
| Jan–Aug 2025 | 14 | 18 | 5 | 13 | 4 | 4 | 3 | 0 | 1 | 0 |
| **Total** | 86 | 84 | 45 | 57 | 32 | 18 | 11 | 5 | 6 | 1 |

*Table 7.* Final Destination of Proposed U.S. AI Legislation (2017–2025)

*Notes:* Average Action Rate: 0.0423 (4.23%). [#]Includes a bill that passed one chamber. [*]Includes two amendments to bills that did pass.

*Table 8.* Bootstrapped Training and Test Accuracy for Each Hurdle Stage

| Stage | Train Accuracy (mean $\pm$ std) | Test Accuracy (mean $\pm$ std) |
|---|---|---|
| Hurdle 1 (any action vs. expired) | $0.7681 \pm 0.0540$ | $0.6455 \pm 0.0779$ |
| Hurdle 2 (advanced vs. stalled) | $0.7702 \pm 0.0518$ | $0.6363 \pm 0.0715$ |

*Table 9.* Mean and Standard Deviation of Annual Action Rates

| Action Rate | Mean (%) | Std (%) |
|---|---|---|
| AI-related bills | 4.23 | 6.32 |
| National congressional | 6.25 | 1.92 |

# E. Reproducibility, Robustness, and Limitations

This subsection documents additional implementation details, variable definitions, robustness checks, and limitations to support reproducibility and clarify the scope of the empirical findings.

## E.1. Data Inclusion Criteria and Query Procedures

**U.S. Congress bills.** Bills were retrieved from Congress.gov by searching for the keywords "Artificial Intelligence" and "AI" and then restricting to bill-level legislative items. Returned items were screened to remove false positives where "AI" referred to non-artificial-intelligence terms or where the bill was not substantively related to AI policy. Duplicates and near-duplicates were deduplicated by bill identifier. Companion bills introduced in both chambers were retained as distinct legislative items because they follow separate procedural paths.

**Global enactments and non-legal actions.** Global enacted policies and non-legal AI safety actions were collected from the sources listed in Appendix A. These are used as a proxy for the pace of binding governance relative to the pace of model capability releases and should not be interpreted as a direct comparison to U.S. congressional procedure.

## E.2. GPT-4o Assisted Sub-field Labeling Protocol

To categorize bills into policy sub-fields, we used a model-assisted labeling pipeline followed by targeted human auditing.

**Prompt inputs.** For each bill, the model was provided (i) the full list of sub-fields with short definitions and (ii) a URL to the Congress.gov bill page.

**Prompt template (abridged).**

> You are assisting with policy categorization. Given the list of sub-fields and the bill URL, return (1) all applicable sub-fields for the bill, allowing multiple labels, and (2) a confidence rating for each label (Low, Medium, High). Output a concise JSON-like list.

**Decision rule.** All labels returned with Medium or High confidence were accepted. Any bill with at least one Low-confidence label was manually audited and corrected by a human annotator. Sub-field definitions were formulated by a subject matter expert and verified by a lawyer.

**Audit procedure and accuracy estimate.** To estimate labeling accuracy, 50 bill-to-sub-field assignments were randomly sampled and manually verified against bill text and summaries, yielding 94% accuracy (47 of 50 correct).

**Reproducibility note.** Because model outputs can vary across runs and model versions, the released dataset will include final human-verified labels and the bill URLs so that results can be reproduced directly from the published annotations.

## E.3. Feature Dictionary for the Logistic Regression

Table 10 defines the outcome and predictors used in the penalized logistic regression.

*Table 10.* Feature Dictionary for the Stalling Model

| Variable | Definition | Encoding |
|---|---|---|
| Hurdle 1 Outcome | Indicator that a bill received any committee action, clearing the engagement gate. | 1 = engaged, 0 = no engagement |
| Hurdle 2 Outcome | Conditional on clearing Hurdle 1, indicator that a bill advanced beyond committee to a resolved outcome. | 1 = advanced, 0 = stalled |
| Num_Sponsors | Total number of sponsors and cosponsors listed on Congress.gov at data collection time. | Integer |
| Bipartisan | Whether at least one cosponsor is from a party different from the sponsor. | 1 = yes, 0 = no |
| Chamber_Binary | Chamber of origin of the bill. | House = 0, Senate = 1 |
| Sponsor_Party_Binary | Sponsor party affiliation. | Dem = 0, Rep = 1 |
| Sub-field indicators | Multi-label sub-field membership (one bill can map to multiple sub-fields). | Multi-hot (0/1) |
| Advanced AI | Aggregated indicator for low-frequency topics (LLM, AGI, Autonomous Driving). | 0/1 |

## E.4. Implementation Notes on Endpoints and the Action Rate

**Endpoints.** Legislative endpoints were assigned through manual annotation based on Congress.gov action histories. A bill was coded as stalled in committee if it had multiple actions but its final recorded action was referral to a committee or subcommittee, followed by session expiration without further progress.

**Action Rate.** The Action Rate is defined as $(P + F)/T$, where $P$ is the number of bills that passed, $F$ is the number of bills that reached a terminal failure outcome, and $T$ is the total number of proposed bills in the dataset window. Bills passing one chamber or passing as amendments are tracked separately and are not counted as enacted bills unless they became law.

## E.5. Robustness Check: Alternate Outcome Definition

To assess whether the primary result is sensitive to endpoint coding, we ran an alternate specification that defines the outcome as *committee-stall only*, excluding "expired without action" and "calendar inaction" categories from the analysis.

### E.5.1. PROCESS DIAGRAM FOR THE TECHNICAL ANALYSIS PIPELINE

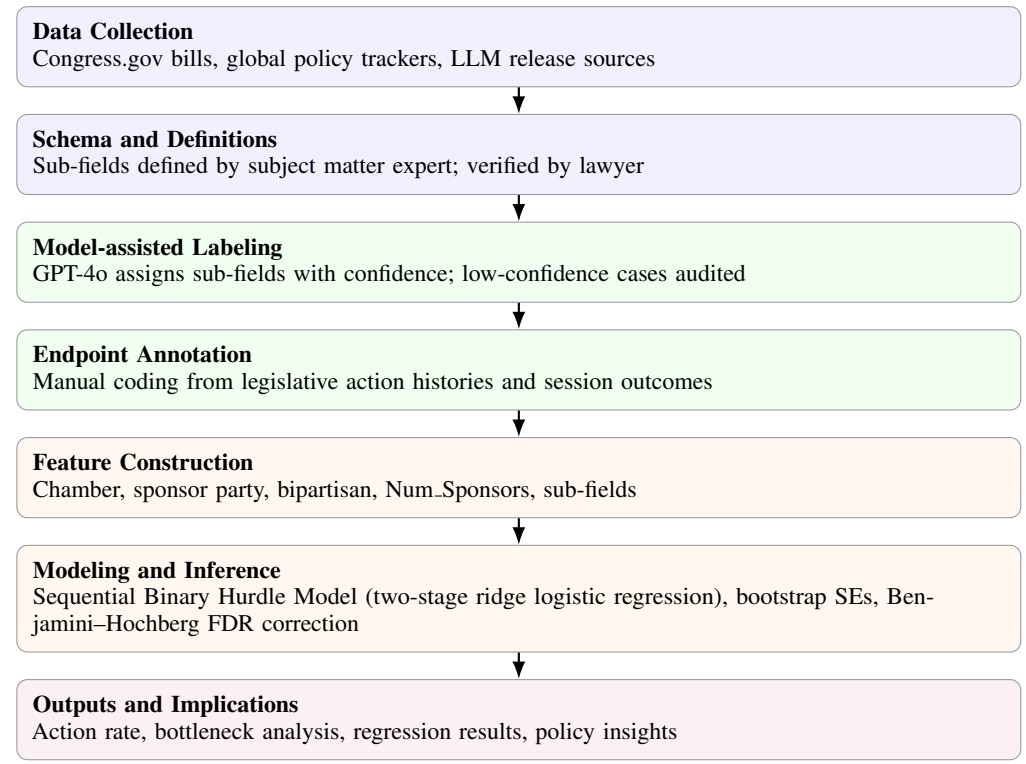

*Figure 4.* End-to-end pipeline for dataset construction, annotation, modeling, and analysis.

## E.6. Limitations and Threats to Validity

- **Correlational findings.** Regression associations should not be interpreted as causal effects, since sponsorship and procedure may be endogenous to expected bill outcomes.

- **Omitted variables.** Committee assignment, bill salience, leadership support, and external events may influence stalling but are not fully captured by the current feature set.

- **Endpoint coding noise.** While endpoints were manually annotated, ambiguous legislative histories may introduce classification error, especially for bills with sparse actions.

- **Model-assisted labeling variance.** GPT-assisted sub-field labels were audited, but residual labeling noise may remain, and results may differ if alternative taxonomies are used.

- **Partial-year data.** The 2025 counts reflect a partial year and may undercount both proposals and enactments relative to full-year totals.

- **Comparability across arenas.** Global enactments are used as a proxy for binding governance pace and are not directly comparable to U.S. congressional outcomes.

- **Generalizability.** Findings about U.S. congressional procedure may not transfer to parliamentary systems or regulatory regimes led by executive agencies.

