# OpenReview forum: "Position: Bridge the AI development-regulation gap through dedicated committees and adaptive legislation"
_ICML.cc/2026/Position_Paper_Track — ICML 2026 Position Paper Track regular_

### Official Review · Reviewer_89Dt · 2026-02-25

**Significance:** 2
**Argument Clarity:** 2
**Rating:** 4
**Confidence:** 4

**Questions:**

- Can you elaborate on what you mean by legislative progress? Should there be national, regional, or global consensus on how AI is developed in your view? How would be establish that given national interest in the case of AI is to be better than "the others".
- Sunset clauses are an interesting approach. Can you expand on that more? This kind of regulation is not widely known but offers a viable way to making laws more adaptive, I think.

**Alternative Views Section:**

Yes

**Compliance With Llm Reviewing Policy A Conservative:**

Affirmed.

**Discussion Potential:**

1

**Final Justification:**

The authors have alleviated my main concerns and the proposed changes have the potential to serve as a foundation for meaningful discussions.

**Paper Summary:**

This position paper argues that the gap between AI development and regulation must urgently be bridged, supported by a quantitative analysis of 150 U.S. Congressional AI bills from 2017–2025, finding that only 4.23% reach any terminal outcome and none of the three enacted bills explicitly address AI safety. The primary cause of failure is procedural rather than substantive: 59% of bills stall at committee stage, and the number of co-sponsors is the only statistically significant predictor of stalling. The paper diagnoses four deeper causes: competitive economic incentives prioritizing speed over safety, institutional inertia, information asymmetries between developers and legislators, and short-term political incentives. It concludes with coordinated calls to action for developers, policymakers, economists, and investors to collectively close the governance gap through built-in alignment, dedicated AI oversight bodies, and preemptive threshold-based regulation.

**Position:**

No

**Position In Title:**

No

**Related Work:**

2

**Strengths And Weaknesses:**

**Strengths:**
- The paper addresses a fundamental challenge that exists in AI-related tasks and beyond. Building solutions with cross-functional teams is an important issue and such initiatives often fail, especially when legislation is too tight.

**Weaknesses:**
- I really appreciate the quantitative support for the arguments in the paper (e.g., the stats on how many bills make it across the finish line, yet I'm not sure what I can take away from Figure 1. How are LLMs and Laws correlated? I mean, the current political environment shows its effects (see esp. the US Executive Order on Trustworthy AI being revoked and a new order on building ).
- The studies talking about the EU AI Act are quite outdated. In 2025, the EU AI Office assumed their work [1, 2]. These organs are exactly what is meant to build the bridge between practitioners and legislators (with a lot of industry influence).
- Overall, the paper reads more like a classical survey than a position paper that opens a lot of opportunity for new discussion. The problem is widely known and an inherent challenge between development/deployment and regulation.
- In section 2, I am not sure how table 2 helps the line of argumentation. It seems most of the factors presented in the paper do not offer enough support for any evidence-based claims. The number of sponsors for a bill is pretty obvious in my eyes (there's also a little mistake: p > 0.05 is bold not the other way around; line 206 right).
- In my view, the alternative position is weak. The point "speed over oversight" is valid and in its current form a potential issue but does speed mean that we cannot have regulation? To what extent does it make sense to have regulation? These are points I would have hoped for to read about in the paper.

[1] EU AI Office, https://digital-strategy.ec.europa.eu/en/policies/ai-office, incl. Continent Action Plan

[2] EU AI Action Plan and Apply AI Strategy, https://digital-strategy.ec.europa.eu/en/policies/european-approach-artificial-intelligence

[3] US Executive Order 14179, https://www.whitehouse.gov/presidential-actions/2025/12/eliminating-state-law-obstruction-of-national-artificial-intelligence-policy/

**Support:**

2

---

> ### Author Rebuttal · Authors · 2026-03-31
>
> We thank Reviewer 89Dt for their interest in sunset clauses and for recognizing that our work diagnoses four deeper causes: competitive economic incentives, institutional inertia, information asymmetries between developers and legislators, and short-term political incentives. We address all concerns below.
>
> **W1: Figure 1**
>
> The key takeaway of Figure 1 is the widening gap between development (accelerated frontier model releases) and governance (flat binding laws). We will clarify this in Sec 2.1
>
> **W2: EU AI Act citations**
>
> Thank you for the citations. We will add the following discussion: Recently adopted Digital Omnibus grants the AI Office more oversight. However, studies note that the absence of redress mechanisms remains a limitation (Leyden 2025; Cabrera et al., 2026).
>
> **W3: Reads like survey, not position**
>
> We take this concern seriously and address it. The current manuscript leads with analysis before making argumentative claims. This is an intentional choice to surface a grounded position on a topic that frequently raises hot debates with starkly different views from stakeholders across frontier AI labs, academics, and legislators.
>
> The analysis-first approach in the introduction may have created the impression of a survey; however, the multi-faceted nature of the topic requires grounding to lead to the right questions. Our paper differs from a survey in that it argues for specific, frequently contested claims: 1) urgent need for regulation by addressing procedural bottlenecks, 2) threshold-based regulation instead of reactive lawmaking, 3) dedicated AI committees instead of existing committees, to reduce pigeonholing, and 4) calls for adaptive governance mechanisms such as sunset clauses.
>
> To mitigate the survey impression, we will:
>
> - Open with our argumentative claim in the introduction and follow with supporting evidence.
> - Sharpen the distinction between findings (what the data shows in Sec 2) and recommendations (what should be done in Sec 4/5), making the position more explicit.
>
> **W4: Table 2/typo**
>
> Table 2 examines how structural and topical features correlate with bill stalling. This is important to identify how enactment gaps can be addressed. The table suggests that structural and political factor (num_sponsors) exhibit a stronger association than topical factors (ex: job security). This finding informs Sec 3, identifying four deeper causes for why AI policies struggle to keep pace. We will make the connection explicit in Sec 2.4.
>
> Typo is fixed.
>
> **W5: speed vs. oversight**
>
> We will further expand the discussion by adding:
>
> >*To What Extent Should AI Be Regulated? Regulation should be stratified by capability and risk. Consumer applications with individual risks should require transparency and consent frameworks such as those in WA state legislation HB2225. Broader-risk systems, such as those capable of generating bioweapons, should require a mandatory pre-deployment safety evaluation. A threshold-based framework should trigger oversight at defined capability benchmarks (dangerous capabilities, deployment scale), to concentrate regulation on riskier systems.*
>
> **Q1: Legislative progress/level**
>
> By "legislative progress" we mean movement from proposal to binding enactment, captured by our action rate metric. It means that system should produce terminal outcomes (pass or fail) rather than indefinite stalling.
>
> On governance level: we favor a layered approach as discussed in our responses to alternative views. National legislation handles jurisdiction-specific implementation. Regional frameworks (like the EU AI Act) set binding floors. International coordination (like the GPAI Code of Practice) establishes minimum standards for catastrophic risks. The goal is interoperability and minimum safety standards, not uniformity.
>
> **Q2: Expand sunset clauses**
>
> We appreciate the interest and will expand to include:
>
> >*Sunset clauses (provisions that cause legislation to automatically expire unless actively renewed) are well-suited to AI governance for three reasons. First, rapid advancements in AI can render regulations obsolete; the expiration due to sunset clauses forces regular reassessment by legislators. Second, sunset clauses lower the initial political cost of enactment by framing new regulation as temporary and reversible, which can reduce opposition from both industry and cautious legislators. Third, they create built-in feedback loops where each renewal cycle generates an intervention for updating based on new technical evidence and stakeholder input.*
>
> >*For AI specifically, we propose the following design: AI safety legislation should include a 5-year sunset with review at 3-year mark. The review would be conducted by the independent AI safety agency we propose to avoid the pigeonholing dynamics we document. Traditionally, sunset clauses are avoided to ensure that legislation isn’t done without future foresight; however, this can be addressed via review mechanisms.*

---

> > ### Author Rebuttal · Reviewer_89Dt · 2026-04-02
> >
> > Thank you for your rebuttal. The explanations helped resolve my concerns and the proposed additions will improve the argument clarity, I think. That said, I believe the paper can be a valuable starting point for a broader discussion.
> >
> > I have one follow-on question related to your response for Q1: How do you envision the "terminal outcomes" to propagate through the layers (e.g., take the case of the EU, just because a national legislation fails in one member state, it could succeed in another. What would that mean for the regional envelope and how do you define success gates)?

---

### Official Review · Reviewer_fF1M · 2026-03-04

**Significance:** 4
**Argument Clarity:** 4
**Rating:** 5
**Confidence:** 4

**Questions:**

1. The number of sponsors is the most significant predictor of legislative stalling, with a coefficient of 0.8068, which indicates that bills with more cosponsors are more likely to stall.  However, in qualitative discussion, you suggest that legislative quality, including coalition building increases the likelihood of enactment. Given that building large coalitions involves increasing the number of cosponsors, how do you reconcile your recommendation with your empirical finding that it is the primary correlate of legislative failure?

2. Your logistic regression model has a test error of 24.0%, and you acknowledge that it omits critical factors (committee assignments, timing, bill salience). In your analysis, you note that industry coalitions are “highly organized” and their lobbying is relentless. How can you be certain that the bottleneck “number of sponsors” is not only a proxy for bill complexity or industry resistance?

**Alternative Views Section:**

Yes

**Compliance With Llm Reviewing Policy A Conservative:**

Affirmed.

**Discussion Potential:**

4

**Paper Summary:**

Contributions:
1. Only 4.23% of U.S. AI bills reach any terminal outcome, lower than the national congressional average of 6.25% over the same period.
2. From penalized logistic regression, found out that structural factors can be used to predict a bill stalling in the committee.
3. Novel dataset and taxonomy of AI policy subfields are created.

Position advocated:
1. Establishment of dedicated AI policy committees, implementation of preemptive enactment models that trigger oversight at defined technical thresholds, and create independent AI Safety Agencies.
2. Treatment of safety and value alignments as first-order engineering constraints rather than post-deployment corrections.
3. The design of market-based safety incentives.

**Position:**

Yes

**Position In Title:**

Yes

**Related Work:**

4

**Strengths And Weaknesses:**

Strengths:
1. Grounded in the 150 U.S. Congressional bills.
2. Action rate helps show AI legislation (4.23%) lags behind the national congressional average (6.25%)
3. Penalized logistic regression helps identify the structural factors and analyze why the problem persists.

Weaknesses:
1. The dataset consists of only 150 bills, which might lead to unstable coefficients for high-interest topics like AGI and LLM, which had only one or two observations each.
2. The regression results, such as the correlation between the number of cosponsors and legislative stalling, are purely correlational and not causal. This makes it unclear if the procedural factors are cause of the gap or only proxies for deeper issues like politics or bill complexity.
3. The omission of critical variables like leadership support, committee assignment, and external geopolitical events weaken the ability to explain the enactment gap.

**Support:**

3

---

> ### Author Rebuttal · Authors · 2026-03-31
>
> We deeply appreciate Reviewer fF1M's accept recommendation and excellent ratings on significance, discussion potential, argument clarity, and related work.
>
> **W1: Small dataset / unstable coefficients for AGI, LLM**
>
> You’re bringing up a well-founded concern. We aggregated low-frequency subfields (LLM: 6 bills, AGI: 5 bills, Autonomous Driving: 1 bill) into a single "Advanced AI" category before running the regression, specifically to reduce unstable coefficients (documented in Appendix D.4). We recognize this was insufficiently highlighted. In the camera-ready, we will: (1) foreground the aggregation rationale in Sec 2.4 rather than leaving it only in the appendix and (2) report bootstrap confidence intervals to make coefficient stability transparent.
>
> We also note that n=150 represents the population of all AI-related bills in this period, not a sample, so the constraint is the phenomenon itself, not sampling design.
>
> **W2: Politics or bill complexity**
>
> We agree that deeper issues such as politics or bill complexity in addition to procedural bottlenecks may influence bill outcomes. In Sec 3.2, we discuss political science literature for all three, 1) institutional factors: e.g. legislative inertia, bureaucratic cost leads to the gap (Cohen, 2021; Kingdon, 1984), 2) complexity: committees with broad interests such as those of Commerce, Judiciary, and Defense, may delay consensus on both the scope and substantive content of proposed legislation, and 3) political issues: such as incentives of agenda setting authorities can slow or deem bills to be of lower urgency, preventing many from reaching floor consideration (ProQuest, 2025; GovTrack, 2004; Wiseman, 2023). We will clarify this more explicitly.
>
> **W3: Omitted critical variables**
>
> We acknowledge this limitation in Appendix D.6. However, to maintain objectivity of our analysis, we constrained our feature-set on information provided by the Congress.gov API in a machine-readable format. Variables such as leadership support, committee assignment, and external geopolitical events require manual coding from hearing transcripts, proprietary lobbying databases and news sources, resources beyond the current scope. Given their manual nature, these variables are expected to introduce labeling uncertainty based on the auditor. We frame these as priority targets for future researchers to expand upon in our dataset.
>
> **Q1: Reconciling coalition-building recommendation with negative Num_Sponsors coefficient**
>
> Our recommendation is to build legislative quality, which includes finding the right amount of coalition partners with influence, instead of increasing the number of co-sponsors. This is driven from political science literature where Sotoudeh et al., 2024 showed that broader cosponsorship by several moderately influential representatives instead of a few influential representatives can reduce the likelihood of advancement through committee stages; and Riker’s 1962 minimum winning coalition theory highlighted rational actors preferring coalitions that are just large enough to win, as broader coalitions diluted payoffs. We will clarify this synthesis in Sec 2.4.
>
> **Q2: Certainty that Num_Sponsors isn't a proxy for complexity or industry resistance**
>
> We cannot be certain from these data alone, and we will state this more prominently. However, our data indicates that it is procedural features (such as num_sponsors) instead of topical features (job security, ethical usage, etc) that correlated with stalling. We note: (1) we control for policy subfields, which should absorb much of the topic-complexity variation; (2) the coefficient persists across robustness checks (Appendix D.5).
>
> Given this, we agree, and based on political science analysis, we hypothesize that Num_Sponsors captures coordination costs, bill complexity, strategic sponsor accumulation on contentious bills, and possibly lobbying intensity. Disentangling these mechanisms requires lobbying expenditure data and committee hearing transcripts, which we identify as the natural next step. Our released dataset is designed to support this future work.

---

> > ### Author Rebuttal · Reviewer_fF1M · 2026-04-01
> >
> > N/A

---

### Official Review · Reviewer_dYpM · 2026-03-12

**Significance:** 3
**Argument Clarity:** 2
**Rating:** 5
**Confidence:** 4

**Questions:**

1. What was the reason for omitting certain variables from the logistic regression analysis?
2. What could be the reason for the number of sponsors to be negatively correlated to advancing the bill?

**Alternative Views Section:**

Yes

**Compliance With Llm Reviewing Policy A Conservative:**

Affirmed.

**Discussion Potential:**

3

**Final Justification:**

See rebuttal acknowledgement.

**Paper Summary:**

This position paper argues that there is an existing gap between the development of AI systems and the development of regulatory frameworks to safeguard them. AI systems undergo rapid advancement, but AI-related safety legislation often lags, and many of them don't get through all the approval stages. The authors show this using a quantitative analysis of 150 AI-related US congressional bills in terms of their subfields, combined with structural and political factors to understand why few proposals ultimately become law and how it results in the regulatory gap. The authors find that many proposed bills focus on issues not related to AI safety, and a logistic regression analysis suggests that the number of bill sponsors is more significantly related to legislative stalling instead of the bill policy itself. Also, the authors identify different procedural bottlenecks (such as pigeonholing) as major contributors to stalling. Finally the authors propose a set of actionable recommendations for different stakeholders to bridge the gap between AI development and its regulation.

**Position:**

Yes

**Position In Title:**

Yes

**Related Work:**

2

**Strengths And Weaknesses:**

**Strengths:**
1. The authors use a data-driven approach, including a logistic regression analysis with significance testing, which provides empirical evidence to support their position.
2. The paper covers an important topic that is becoming more relevant as AI gets embedded in multiple aspects of our daily lives.
3. The additional contribution of curating a dataset can help foster future research on this topic.
4. The authors consider multiple stakeholders in their call for action, which is important for large-scale regulatory changes.

**Weaknesses:**
1. The paper does not include a discussion about the growing developments in AI safety research, including safety benchmarks, interpretability methods, or red-teaming frameworks. How these advancements impact the regulatory efforts is an important consideration.
2. The authors suggest that there should be coordinated regulatory efforts across different nations, but the argument for it seems a bit weak. Nations often compete for technological leadership and economic advantages in AI so it doesn't seem feasible to have such a coordinated effort.
3. Some of the proposed calls to action for developers seem to be difficult to operationalize:
- Integrating alignment objectives directly into system design seems conceptually valid, but it is challenging to implement for large-scale generative models with trillions of parameters. Very few organizations have the resources to train such systems, and changing pretraining objectives incurs high costs.
- The suggestion to open-source models and datasets is appealing, but it can conflict with the intellectual property and competitive considerations of the commercial AI developers. For instance, many such organizations that started by open-sourcing their models have now switched to closed-source.
- The recommendation for developers to engage directly with legislators may not be realistic in many cases as it needs speacialized legal, regulatory, and communication expertise.
4. It is surprising that the logistic regression model achieves higher testing accuracy compared to the training accuracy, especially given the small dataset. This leads me to believe that the model might be underfitting. Also, the test accuracy is not too high, which makes it difficult to draw strong conclusions about feature importance.
5. There are a few typos in the paper: "safegaurd" -> "safeguard" (line 106), "compress" -> "compressed" (line 266), "Table ??" (line 999), the sentence "While technological changes ..." is grammatically incorrect

**Support:**

3

---

> ### Author Rebuttal · Authors · 2026-03-31
>
> We thank Reviewer dYpM for the thorough review and for recognizing our data-driven approach to support the position, dataset contribution, and multi-stakeholder call-for-action, which is needed for large-scale regulatory changes. We address all remaining concerns with concrete revisions.
>
> **W1: Missing AI safety research context**
>
> Thank you for this excellent point. We will add the following contextualization to Sec 3:
>
> >*"While technical AI safety research is advancing, including evaluation benchmarks such as HELM (Liang et al., 2023), FOL AI Safety Index (Future of Life, 2025), DecodingTrust (Wang el al.,2023), mechanistic interpretability methods such as Anthropic's 2025 circuit tracing and structured red-teaming protocols from NIST, these remain voluntary practices rather than binding regulatory requirements. These tools can additionally supplement regulatory frameworks."*
>
> **W2: International coordination feasibility**
>
> We agree that international coordination is challenging. However, as noted in Sec 1, nations have historically coordinated despite strong economic incentives to defect. For the Montreal Protocol, 197 nations aligned despite initial resistance from major companies such as DuPont, which actively opposed regulations to protect profits. Recently, highly competitive firms started to align on safety standards. The EU GPAI Code of Practice (2025) was signed by Amazon, Google, Microsoft, OpenAI, and Anthropic, all direct competitors who nonetheless agreed to shared oversight provisions. These indicate that international alignment in AI is also feasible. Additionally, we advocate minimum standards (analogous to the Basel Accords in banking).
>
> **W3: Operationalizing developer recommendations**
>
> 1) ***Built-in alignment:*** Agree we will revise the text to include evaluation stage: "Integrate alignment evaluation as a first-order engineering gate, through safety cases, red-teaming, and pre-deployment review, rather than treating it as a post-deployment correction."
>
> 2) ***Open-sourcing:*** We do not advocate mandatory open-sourcing and will reframe it: "Where full open-sourcing is not feasible, developers should reduce information asymmetries through structured transparency (including model cards, safety evaluation disclosures, and research API access)."
>
> 3) ***Legislator engagement:*** Yes, this is directed at organizations with policy capacity. Revised text: "Proactively engage with policymakers through organizational policy teams and industry associations."
>
> **W4: Model underfitting**
>
> We thank the reviewer for the observation. All inferential results in the manuscript (coefficients and their interpretation) were derived using bootstrap resampling. However, the discrepancy (test > train) in the Appendix arose from accuracy based on a single train-test split, which is sensitive to sampling variability.
>
> To ensure consistency, we now report bootstrap-based estimates for model accuracy in the Appendix. Under the bootstrap, training accuracy is 0.775 (95% CI: [0.658, 0.873]) and testing accuracy is 0.726 (95% CI: [0.579, 0.840]). This indicates that the earlier difference was due to split-specific variance.
>
> Predictive accuracy: As our objective is inferential, we interpret the model using bootstrap resampling, focusing on whether coefficient signs and magnitudes remain consistent across resamples. The bootstrap results indicate stable directional effects, suggesting identified associations are not driven by a single split or noise. We will clarify this in Sec 2.4.
>
> **W5: Typos**
>
> Corrected.
>
> **Q1: Omitted variables**
>
> To maintain objectivity of our analysis, we constrained our feature set on what the Congress.gov API provides as metadata. Variables such as leadership support, bill salience, and lobbying intensity require manual coding from hearing transcripts, news sources, and proprietary lobbying databases, resources beyond the current scope. The manual coding process risks introducing noise due to auditor subjectivity. We frame these as targets for future researchers to expand on our dataset.
>
> **Q2: Why Num_Sponsors negatively correlates with advancement**
>
> As discussed in Sec 2.4, political science literature offers several explanations. Sotoudeh et al., 2024 show that broader cosponsorship by several moderately influential representatives instead of a few influential representatives can reduce the likelihood of advancement through committee stages. Kessler & Krehbiel (1996) show that cosponsorship can serve as a strategic signal without guaranteeing passage. This is in line with Riker’s (1962) minimum winning coalition theory, which argues that rational actors prefer coalitions that are just large enough to win, as larger coalitions dilute payoffs. For AI Bills in the US Congress, a high sponsor count likely proxies for a signaling statement or slowdown of the legislative process, instead of optimizing for bill passage. We will further clarify this interpretive context in Sec 2.4.

---

> > ### Author Rebuttal · Reviewer_dYpM · 2026-04-02
> >
> > I thank the authors for their response to my questions and concerns. I will updated my score accordingly. Looking at the other reviews, I agree that a restructuring is needed to establish this as a position paper and to garner discussion, so I hope the authors incorporate the points discussed during the rebuttal.

---

### Official Review · Reviewer_giAw · 2026-03-12

**Significance:** 3
**Argument Clarity:** 3
**Rating:** 5
**Confidence:** 3

**Questions:**

- Part of motivation comes from comparing the number of LLM models with the number of pieces of legislation. I'm not sure that's a good comparison. How do you count the impact of the EU directive that affects 27 countries?

**Alternative Views Section:**

Yes

**Compliance With Llm Reviewing Policy A Conservative:**

Affirmed.

**Discussion Potential:**

2

**Final Justification:**

I think the rebuttal has addressed most of my concerns. The authors have updated their analysis and improved it significantly. My one remaining concern is with respect to discussion potential, but I think it's good enough to accept.

**Paper Summary:**

The paper suggests that legislation on AI concerns have lagged behind development of AI tools and identifies that as a problem as well as proposes a set of recommendations for various stakeholders in order to improve the situation.

**Position:**

Yes

**Position In Title:**

Yes

**Related Work:**

3

**Strengths And Weaknesses:**

## Strengths

- The topic is timely, the problem is real, and most would agree that this is a problem.
- The position is motivated by ample reasoning, literature, and empirical work

## Weaknesses

- I am worried that this may not spark that much discussion. I think there is wide agreement that regulation and legislation need to keep pace with the development of AI, but the call to action is quite loose and too overreaching to, I think, provide good grounds for discussion. It would help if it was more focused.
- The statistical analysis used to motivate the topic is not that robust. Without a baseline for how many bills stall or fail, how are we to gauge whether this is a specific problem for AI legislation? It is also quite problematic to even dichotomize the outcome as you do. Furthermore, the Bonferroni correction is typically thought to be over-conservative.
- The focus on the US weakens the potential of discussion for the paper. A considerable amount of focus is spent on this example and part of the call to action seems tailored to a US context.

**Support:**

3

---

> ### Author Rebuttal · Authors · 2026-03-31
>
> We thank Reviewer giAw for recognizing the timeliness, empirical work, and argument clarity of our work. We address all concerns below and welcome further feedback.
>
>
> **W1: Limited discussion potential / loose call to action**
>
> We highlight the paper's specific empirical findings and proposals that aim to generate substantive debate:
> We show that despite widespread concern about AI risks, the AI bills (4.23% action rate) underperform the general congressional baseline (6.25%). This quantifies the paradox, inviting discussion about why the legislative gap exists and what makes AI uniquely difficult to regulate.
>
> Our finding that procedural factors (Num_Sponsors) negatively correlate with legislative success (β=0.8068, p<0.01, Table 2) instead of topical features (such as job security, ethical usage) directly indicates that procedural, legislative, and political bottlenecks should be prioritized to achieve AI governance.
>
> As Reviewer dYpM notes, our calls for action are structured for a multi-stakeholder framing, which is important for large-scale regulatory changes. The proposals identify specific actions to overcome procedural bottlenecks and raise critical questions for debate:
>
> - The proposal for threshold-based regulation raises contested questions: At what compute level should oversight activate? Who sets the thresholds? Should there be deployment restrictions?
>
> - The call for dedicated AI committees, motivated by our 59% pigeonholing finding, raises questions about jurisdictional redesign, as creating new standing committees is often resisted by existing committees on other tech adjacent topics.
>
> - Our call for built-in rather than post-deployment alignment is a recommendation that is contested in the alternative views Sec (The Afterthought Argument). Developers of AI systems place alignment at different stages of system development; while some contest that post-hoc alignment is sufficient, others argue that systems should be designed with built-in alignment objectives. We support the latter.
>
> We believe these specific findings and proposals provide ample ground for discussion. We will ensure to clarify these in the Camera Ready version.
>
> **W2: Statistical robustness concerns**
>
> ***Baseline:*** We note that our paper provides a direct baseline comparison in the abstract and Sec 3.2: the national congressional average action rate is 6.25% over the same 2017-2025 period, against which our 4.23% AI-specific rate demonstrates underperformance. We will make this comparison more prominent in the camera-ready version.
>
> ***Dichotomization:*** We agree this simplifies outcomes. However, Appendix D.5 documents our robustness check with an alternate outcome definition (committee-stall only). The binary encoding enables consistent comparison across years, and our released dataset preserves the full endpoint taxonomy (seven categories) to support future survival analysis or multinomial modeling by other researchers.
>
> ***Bonferroni:*** We agree it is conservative and chose it intentionally to guard against false positives given our multiple comparisons. The fact that Num_Sponsors survives this stringent correction strengthens our confidence in the finding.
>
>
> **W3: US-centric focus**
>
> We appreciate this concern and agree that broader geographic coverage would strengthen the paper's reach. Our choice of the US as the primary empirical case is motivated by a specific paradox: the world's leading developer of frontier AI models has the weakest binding regulatory framework among major jurisdictions (Table 1). We believe understanding why this paradox exists is critical to shaping guidelines globally.
>
> That said, our procedural findings (committee bottlenecks, coordination costs from multi-sponsor dynamics, pigeonholing) generalize to any fragmented or federal governance system. Table 1 situates US findings within the global context, and we will incorporate the EU AI Office developments (per Reviewer 89Dt's helpful citations) in the camera-ready version to strengthen the comparative framing.
>
>
> **Q1: LLM vs. legislation comparison / EU directive impact**
>
> We agree that raw counts create an asymmetry and that the EU AI Act's impact across 27 member states is not captured by counting it as one enactment. To clarify: Figure 1 is intended to illustrate temporal pace mismatch (model releases accelerating while enactments remain flat), not a 1:1 scalar correspondence. Even if the EU AI Act were weighted as 27 units, the rate-of-change divergence would persist. We will revise the figure caption to read: "Figure 1 demonstrates temporal acceleration asymmetry: frontier LLM releases have accelerated substantially since 2022, while binding enactments have grown modestly, illustrating a widening governance gap. Counts reflect discrete policy events, not jurisdictional reach."

---

> > ### Author Rebuttal · Reviewer_giAw · 2026-04-01
> >
> > Thatnks for the detailed rebuttal! i have a few follow-up remarks. But I would also have appreciated if you were more forthcoming with considering to modify your paper.
> >
> > > Dichotomization: We agree this simplifies outcomes. However, Appendix D.5 documents our robustness check with an alternate outcome definition (committee-stall only). The binary encoding enables consistent comparison across years, and our released dataset preserves the full endpoint taxonomy (seven categories) to support future survival analysis or multinomial modeling by other researchers.
> >
> > You loose too much power when dichotomizing, and you can still run predictions on top our a continuous outcome after you've fit your model, so you're not helping consistency.
> >
> > > Bonferroni: We agree it is conservative and chose it intentionally to guard against false positives given our multiple comparisons. The fact that Num_Sponsors survives this stringent correction strengthens our confidence in the finding.
> >
> > Sure, but you risk the *opposite* problem: missing important predictors because you're too conservative. Please reconsider.

---

### Decision · Program_Chairs · 2026-04-30

**Decision:**

Accept (regular)

**Comment:**

This paper was supported by all reviewers. On the whole, the novel analysis of Senate bills being passed was appreciated, and ultimately, it was felt that this paper was likely to encourage debate.

While the official position of the paper "Bridge the Gaps between AI Development and Regulation" is unlikely to spark debate, the analysis showing that the US Senate can not pass most bills and is unable in its current form to "Bridge the Gap" is likely to spark discussion. Many of the follow-up points by the authors explicitly address Senate issues.

The authors should ensure they are clear that this result only holds at the federal level, and is not a result regarding the US as a whole. Figure captions, paper headings, and the response to reviewers tended to play this down.

If the authors intend to report the 4% vs. 6% difference of AI vs general bill in the final version of the paper, they should include the standard deviations.